# Estimation of Fruit Load in Australian Mango Orchards Using Machine Vision

**Nicholas Todd Anderson** [1,*], **Kerry Brian Walsh** [1], **Anand Koirala** [1], **Zhenglin Wang** [1], **Marcelo Henrique Amaral** [1], **Geoff Robert Dickinson** [2], **Priyakant Sinha** [3] and **Andrew James Robson** [3]

1   Institute for Future Farming Systems, Central Queensland University, Rockhampton 4701, Australia; k.walsh@cqu.edu.au (K.B.W.); a.koirala@cqu.edu.au (A.K.); z.wang@cqu.edu.au (Z.W.); m.m.amaral@cqumail.com (M.H.A.)
2   Department of Agriculture and Fisheries, Queensland Government, Brisbane 4880, Australia; Geoff.Dickinson@daf.qld.gov.au
3   Applied Agricultural Remote Sensing Centre (AARSC), University of New England, Armidale 2351, Australia; psinha2@une.edu.au (P.S.); arobson7@une.edu.au (A.J.R.)
*   Correspondence: nicholas.anderson@cqumail.com

**Abstract:** The performance of a multi-view machine vision method was documented at an orchard level, relative to packhouse count. High repeatability was achieved in night-time imaging, with an absolute percentage error of 2% or less. Canopy architecture impacted performance, with reasonable estimates achieved on hedge, single leader and conventional systems (3.4, 5.0, and 8.2 average percentage error, respectively) while fruit load of trellised orchards was over-estimated (at 25.2 average percentage error). Yield estimations were made for multiple orchards via: (i) human count of fruit load on ~5% of trees (FARM), (ii) human count of 18 trees randomly selected within three NDVI stratifications (CAL), (iii) multi-view counts (MV-Raw) and (iv) multi-view corrected for occluded fruit using manual counts of CAL trees (MV-CAL). Across the nine orchards for which results for all methods were available, the FARM, CAL, MV-Raw and MV-CAL methods achieved an average percentage error on packhouse counts of 26, 13, 11 and 17%, with SD of 11, 8, 11 and 9%, respectively, in the 2019–2020 season. The absolute percentage error of the MV-Raw estimates was 10% or less in 15 of the 20 orchards assessed. Greater error in load estimation occurred in the 2020–2021 season due to the time-spread of flowering. Use cases for the tree level data on fruit load was explored in context of fruit load density maps to inform early harvesting and to interpret crop damage, and tree frequency distributions based on fruit load per tree.

**Keywords:** deep learning; fruit-culture; fruit load; precision horticulture; machine vision; prediction quantification; yield estimation

## 1. Introduction

A timely and accurate pre-harvest estimation of crop load is necessary to inform harvest, storage and transport logistics and effective marketing. A forward estimation of crop load is particularly important for crops such as mango (*Mangifera indica* L.) given the short length of time between physiological maturity and on-tree ripening, and the relatively short postharvest life of the harvested product. Australian mango farm managers seek an estimate within ±10% of the actual harvest yield for planning of harvest resourcing and marketing. In current commercial practice, yield is estimated by extrapolation from a manual count of fruit on a sample of trees. However, manual counting is labor intensive, with consequent management pressure to reduce time spent counting each tree and number of trees counted.

In-field counting of fruit on tree using machine vision is possible using imagery collected from RGB cameras mounted on ground-based vehicles [1–5], based on the implementation of deep learning in tree fruit detection, as reviewed by Koirala et al. [6]. Briefly,

the first report of use of machine vision for estimation of mango fruit load [7] employed color and shape thresholding for mango fruit detection, achieving a Percentage Error (PE) of 30 to 44% on manually counted images. More recent studies have employed deep learning architectures, e.g., a Region-based Convolutional Neural Networks (RCNN) [2] and a You Only Look Once (YOLO) algorithm [4] were implemented using imagery of the same mango orchard. Under appropriate imaging conditions, detection of mango fruit within images of whole canopies is now reliable, e.g., F1 score of 0.968 and average precision of 0.983 reported for count of mango fruit per image frame [4].

Two approaches have been taken in scaling from a fruit count per image to a count per tree. In the so-called dual-view method, the sum of fruit detections in two images per tree, one from each inter-row, are adjusted using an occluded fruit factor estimated from a manual count of a sample of trees. In the so-called multi-view approach, a video is acquired from a camera passed down the inter-rows of an orchard, providing multiple viewpoints of each tree side [2,8,9]. Detected fruit are tracked between frames and added to a cumulative fruit count only when not present in a predicted position for several frames. This approach allows a greater proportion of, and possibly all, fruit per tree to be detected compared to the dual-view method.

In a recent review, Anderson et al. [5] recommended that attention should turn from documentation of the effectiveness of the fruit detection algorithms at an image and tree level to focus on estimations at a whole orchard level, with consideration of factors that affect performance. Likely factors include lighting conditions, camera orientation to the canopy and canopy architecture/foliage density. Direct sunlight can create difficult imaging conditions when imaging whole orchards, given the multiple orientation of the camera with respect to the sun. Various approaches to this issue have been reported, including use of an 'over the row' shade for consistent imaging of fruit on apple trees [1], use of intense strobe lighting and short exposure times [2] and imaging at night with use of artificial lighting [4,7].

The orientation of the camera to the canopy should allow camera view of all fruit on trees, or at least of a consistent proportion of fruit on trees. For example, Gongal et al. [1] utilized vertical arrays of cameras under an 'over the row' shade for consistent imaging fruit on apple trees. Upward facing cameras mounted to a tractor bar were used for kiwi fruit yield estimations [10]. For 'modern' mango orchards employing trees to 4 m height and a typical row spacing of 8 m, our research group has described use of a single camera to view the whole tree from an inter-row position, with two cameras at 180° to each other and perpendicular to the row used to imaging both row sides from a vehicle moving through the inter-row [2,4,7].

Any machine vision method is compromised if fruit are not visible from the camera positions because of occlusion by foliage or other fruit, or if fruit are visible from both inter-rows, resulting in double counting. For example, the difference in performance of a machine vision count of two kiwifruit orchards relative to packhouse counts, at 6 and 15% PE, was attributed to canopy density [10]. Anticipating performance in relation to canopy architecture and foliage density is therefore relevant to practical use of the machine vision methods.

The performance of fruit load estimates can be expressed in terms of Absolute Percentage Error (APE) of the estimate relative to a reference measurement. For example, an APE of 2.2–84.3% on harvest yields between 1.1–41.1 t ha$^{-1}$ was reported in estimation of yield of 15 mango orchards using a model based on tree height, canopy volume and a visual assessment of a 'load index' (to categories of low, medium, and high) Sarron et al. [11]. An APE of 6 and 15% was achieved relative to packhouse counts for machine vision based estimates of two kiwifruit orchards [10]. An APE of between 5 and 15% across five mango orchards was achieved using dual-view imaging coupled with an occluded fruit factor, relative to packhouse counts [4]. A Mean Absolute Percentage Error (MAPE) of 18% was achieved for machine vision fruit count of 20 apple trees, compared to harvest

counts [1]. Reported estimates have thus failed to consistently meet a specification of 10% APE, relative to packhouse estimates.

In previous publications, our group has documented the creation of a modified YOLOv3 detection algorithm [4], with implementation of fruit tracking to enable multi-view estimation [9] and development of an imaging system mounted to a ground vehicle, e.g., [12]. This hardware has been used in conjunction with the dual-view method in estimation of yield mango orchards [12,13]. In the current study we report on the performance of this hardware in conjunction with the multi-view method, and act on the recommendation of a review of yield forecast methods [5] for reporting of estimates at an orchard level. Consideration is given to machine vision operating factors such as ambient lighting and the effect of different canopy architectures. Fruit load estimates of up to 20 mango orchards were made in each of two seasons, with comparison to packhouse counts and manual count estimates. This represents the most comprehensive report of whole orchard fruit load estimation available to date, with the work intended to assist efforts to implement these technologies into farm use. Use cases for the orchard fruit load density map produced through the machine vision method are also presented.

## 2. Materials and Methods

### 2.1. Orchards

Data was sourced from 37 *Mangifera indica* (L.) orchards in the 2019–2020 season and 19 orchards in the 2020–2021 season. Orchards were located throughout the major mango producing areas of Australia, and varied in cultivar (Kensington Pride, Calypso®, Honey Gold, Keitt, R2E2 and NMBP1243), tree age and orchard layout, extending the localities of a previous study [12] (see Table A1 of Appendix B). An orchard was defined by farm management, i.e., by areas of consistent management practice (planting date, cultivar, irrigation, nutrition, pruning, pest and disease management), and typically consisted of around 1000 trees.

Where available, orchard harvest output as number of fruits was obtained from the pack-line data management system as tray number and fruit per tray for shipped fruit, and weight of reject fruit divided by estimated average fruit weight. If only weight data, not tray number, was available from the packhouse, fruit number was calculated using an average fruit weight provided by each orchard. Packhouse data will vary from field fruit numbers where: (i) there is fruit left on tree or ground at harvest; (ii) there is mixing of lots during harvest or packing, e.g., a wash tank that is not fully emptied between lots; (iii) when the estimate of average fruit size is incorrect. Several orchard data sets were rejected from the current study when the amount of fruit left in field and/or lot mixing discrepancies in the packhouse were observed to be large.

The mango planting systems trial at the Department of Agriculture and Fisheries, Queensland Walkamin Research Facility [14] was also accessed. This trial consists of four training systems (conventional, hedge, single leader, trellis), of three cultivars (NMBP1243, Calypso®, Keitt), planted to low, medium and high densities (208, 417, 1250 trees/ha). All trees were harvested individually, providing fruit load data per tree. In the machine vision work, counts were made of fruit on replicates of groups of five consecutive treatment trees, with 47 conventional, 26 hedge, 26 single leader and 25 trellis replicates. For cultivars there were 27 NMBP1243, 37 Calypso® and 35 Keitt replicates and for densities there were 22 low, 26 medium and 26 high replicates. Replicate numbers are unequal due to the design of the high density trial [14].

### 2.2. Manual Estimates of Orchard Fruit Load

Manual count estimates of fruit load per tree in commercial orchards were made using hand tally counters by human operators moving around the canopy in a single direction. These estimates were performed after stone hardening stage of fruit development, i.e., after fruit drop has largely ceased, approximately six weeks before harvest. Estimates of fruit load made by farm management (FARM) were obtained for orchards 1–21 in 2019–2020 and

1–16, 18–21 and 24 in 2020–2021. The method used for this estimate varied between farms, with the best practice being a manual count of 25 trees by each of two people per orchard, i.e., a count of 50 trees, or approximately 5% of trees. These trees were typically sampled using a pseudo-systematic sampling approach involving a single transect sampling line through the orchard.

Manual counts were also made for another 18 'calibration' trees per orchard, with the average count of counted trees multiplied by the total number of trees in the orchard to yield an orchard estimate (CAL). Trees were selected as described by Rahman et al. [15]. Briefly, a stratified sampling approach was adopted, with six trees randomly selected from each of three normalized difference vegetation index (NDVI) classes representing high, medium and low vigor trees, derived from WV3 satellite images acquired at the stone hardening stage. Counts per tree were made at least in duplicate and the average count used. The CAL tree counts were also used to estimate a correction factor for occluded fruit in machine vision estimates, calculated as the ratio of the tree count to the machine vision count for each orchard.

At the Walkamin trial site, fruit per tree were counted during harvest, with fruit at harvest maturity stage.

### 2.3. Machine Vision System

An in-field machine vision system developed in-house was used, with a full descriptions of the system in [4,9]. The imaging platform was driven through the orchard at 5–7 km/h. The per area time to image was ~7 min/ha, varying with orchard and headland layout. Imaging was undertaken within days of the manual counts, except for orchards 1–19 of the 2020–2021 season, for which manual count occurred 22 d after imaging.

Briefly, the system consisted of a frame mounted to a farm vehicle, carrying two 500 W LED light panels and two 5 Mp Basler acA2440 RGB camera (Basler, Ahrensburg, Germany) fitted with 5 mm focal length Kowa lens, facing to each row side, associated hardware for image recording and a Global Navigation Satellite System (GNSS; Trimble, Leica GS14 or Emlid v2) (Figure 1). The GNSS system was used to provide a geolocation datum for every image frame. The MangoYOLO convolutional neural network described by Koirala et al. [4] was applied to detect fruit in each frame. Typical camera to canopy distance was 2 m. In the multi-view method, video was acquired at 10 fps. A Kalman filter was adopted to track the fruit across a sequence of frames, following the procedures of [2,9]. The fruit was added to the count when the fruit did not reappear in seven subsequent frames. A fruit size threshold ($12 \times 12$ pixels) was utilized to reduce the error of double counting of fruit seen from both sides of the canopy, with distant fruit appearing smaller in image and thus excluded from count. The sum of the manual fruit count for 18 trees in each orchard was regressed to the MangoYOLO pipeline count of fruit in the two images for these trees to estimate an 'occlusion factor' for each orchard. This correction factor was then applied to the machine vision count of all other trees, following the approach of Koirala et al. [16].

### 2.4. Experimental Exercises

Several exercises were undertaken to document machine vision method performance: (i) an orchard was imaged repetitively through a day to assess multi-view method repeatability and impact of ambient lighting; (ii) multi-view fruit load estimates were acquired for trees of different canopy architectures to enable consideration of the impact of canopy architecture on estimate accuracy; (iii) the impact of reduced sampling of rows for machine vision estimations of orchards was assessed; (v) the impact of an extended flowering period, and thus extended harvest maturity period, on a machine vision load estimate was documented; (vi) fruit load per orchard was estimated using several methods for approximately 20 orchards in each of two seasons. The methods included the farm management estimate, based on average fruit count of trees in a single transect per orchard (FARM), the average of the manual count of 18 trees based on a NDVI stratification multiplied by tree number

per orchard (CAL), and multi-view machine vision, without (MW-Raw) or with correction using an occlusion factor (MV-CAL).

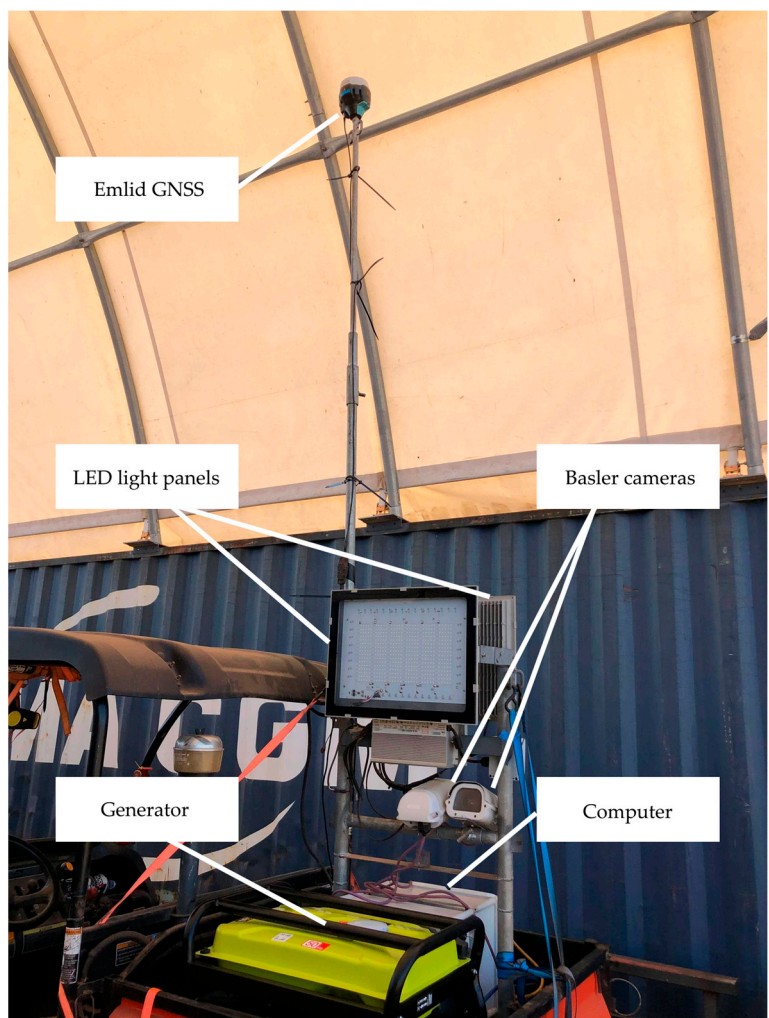

**Figure 1.** Imaging system, featuring GNSS receiver, LED light panels, two Basler cameras, generator and computer.

Finally, several use cases relevant to orchard management were presented based on machine vision data. These cases include a frequency chart for fruit load per tree for an orchard and mapping of fruit load across an orchard.

*2.5. Statistics*

Percentage error (PE) on orchard fruit load estimates was calculated as:

$$PE = \frac{\text{Estimated fruit number} - \text{Packhouse fruit number}}{\text{Packhouse fruit number}} \times 100 \qquad (1)$$

while Absolute Percentage Error (APE) was calculated using the absolute difference of estimate and reference values.

To determine if there were significant differences between the estimation techniques employed, the APE of estimates to packhouse counts for all orchards estimated were tested for Least Significant Difference (LSD) with a Bonferroni p-adjustment to $p < 0.05$ in R package 'agricolae' (version 1.3.3) [17].

## 3. Results and Discussion

### 3.1. Machine Vision Method Validation

3.1.1. Repeatability and Time of Day

In a test of method repeatability, two rows of orchard 30 were imaged twice in the same night. Fruit count for these rows were recorded at 12,902 in the first pass and 13,008 in the second pass, a difference of 106 fruit (0.1% of total). Repeated estimates of orchard 28 were also made through a day on which sunrise occurred at 05:04 and sunset at 18:30. The 08:00, 10:00, 12:00, 15:00, 16:00, 18:00, 20:00 and 21:30 h counts were 37, 49, 47, 54, 53, 48, 98 and 102% of the count at 21:00 h (44,419 fruits). The night repeatability results were equivalent to that achieved by [2,10]. The poor daytime result was ascribed to a higher dynamic range of lighting under sunlight than artificial lighting, and over-exposure when the sun was imaged due to low sun angles in morning and afternoon. Further, the MangoYOLO model had been trained on images collected at night using LED floodlighting, with fruit differentiated from foliage more by intensity than color [4].

The current multi-view imaging system thus achieves good repeatability but is restricted to night-time use. Repeatability has rarely been reported in past studies, but it is recommended as a performance criterion to be included in future reports.

3.1.2. Impact of Canopy Management on Machine Vision Estimation

Stein et al. [2] remarked on the uncertainty of the stability of a ratio of visible to total fruit between trees of an orchard and recommended use of the multi-view over the dual-view method on the basis that multi-view was not dependent on use of an occluded fruit correction. In the mango orchard in which their work was based, the multi-view method achieved view of all fruit on tree, and they called for future work to validate this in other orchards. The current study addresses this call.

Two types of error in multi-view estimation of fruit load varied with canopy type: (i) failure to observe fruit due to occlusion by foliage; (ii) double counting if fruit were seen from the other side of the canopy but not sufficiently differentiated by image size to be rejected by the image size filter.

Multi-view machine vision estimates were compared to harvest counts for trees grown to four canopy architectures at the Walkamin Research Facility. In the trellised system (Figure 2), fruit were seen from both sides of the canopy without differentiation by image size, causing a machine vision overcount (Figure 3, top panel). The NMBP1243 tree over-estimate was most severe in the high-density planting systems (e.g., PE of 32 and 47% for hedge and trellised systems, respectively; see Appendix A). The trellis treatment was therefore removed from the statistical analyses of cultivar and density treatments (Figure 3, middle and bottom panel). For the other canopy systems, tree density did not significantly impact multi-view results (Figure 3, middle panel), although variation in PE was larger in the high than in lower density systems (higher SD, Figure 3, middle panel). This variation was ascribed to variation in foliage density, and thus variation in level of fruit occlusion. PE was significantly different for the three cultivars, with best results for the Keitt trees, which have a relative open canopy structure, followed by Calypso® and NMBP1243 cultivars (Figure 3, bottom panel).

The multi-view technique is therefore recommended for fruit load estimation of orchards using the canopy management systems of conventional, hedge and single leader, but not trellised canopies.

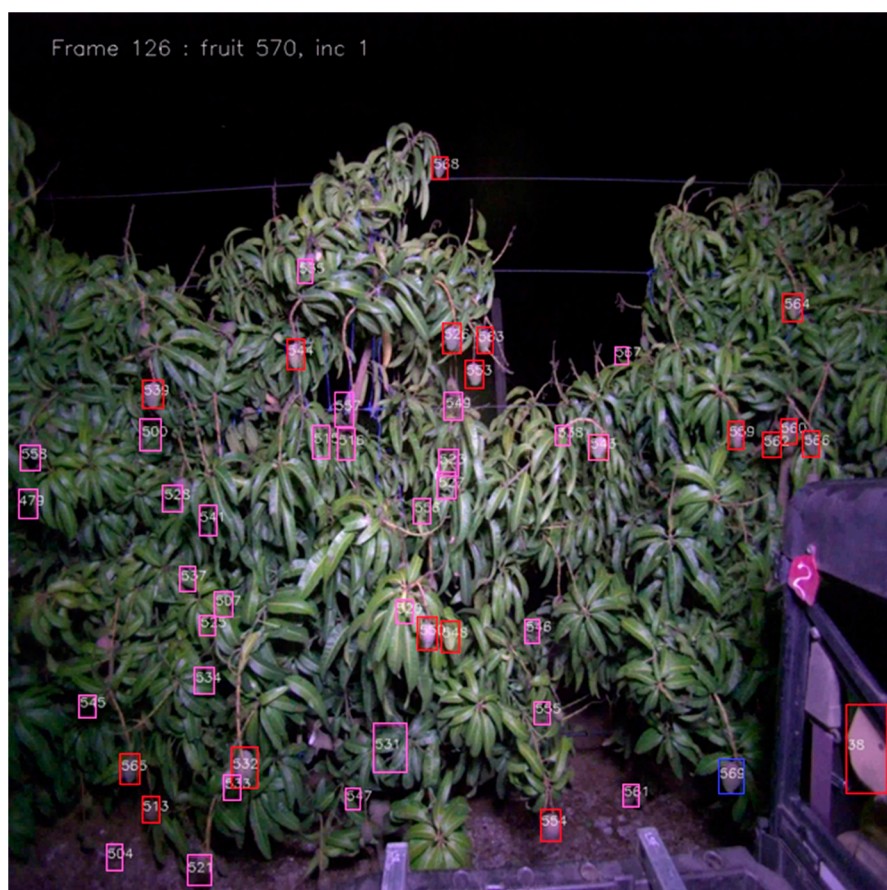

**Figure 2.** Image of trellised NMBP1243 canopy at Walkamin Research Facility. The MangoYOLO algorithm was used to classify and localize mango fruits in image. Blue bounding boxes represent a first detection on the current frame, red bounding box represents a fruit detected and tracked from previous frames and a pink bounding box is a tracking prediction of where a previous fruit detected could reappear.

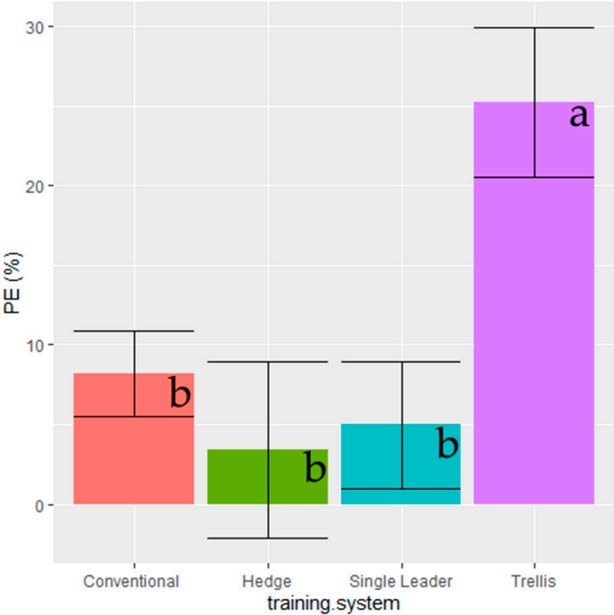

**Figure 3.** *Cont.*

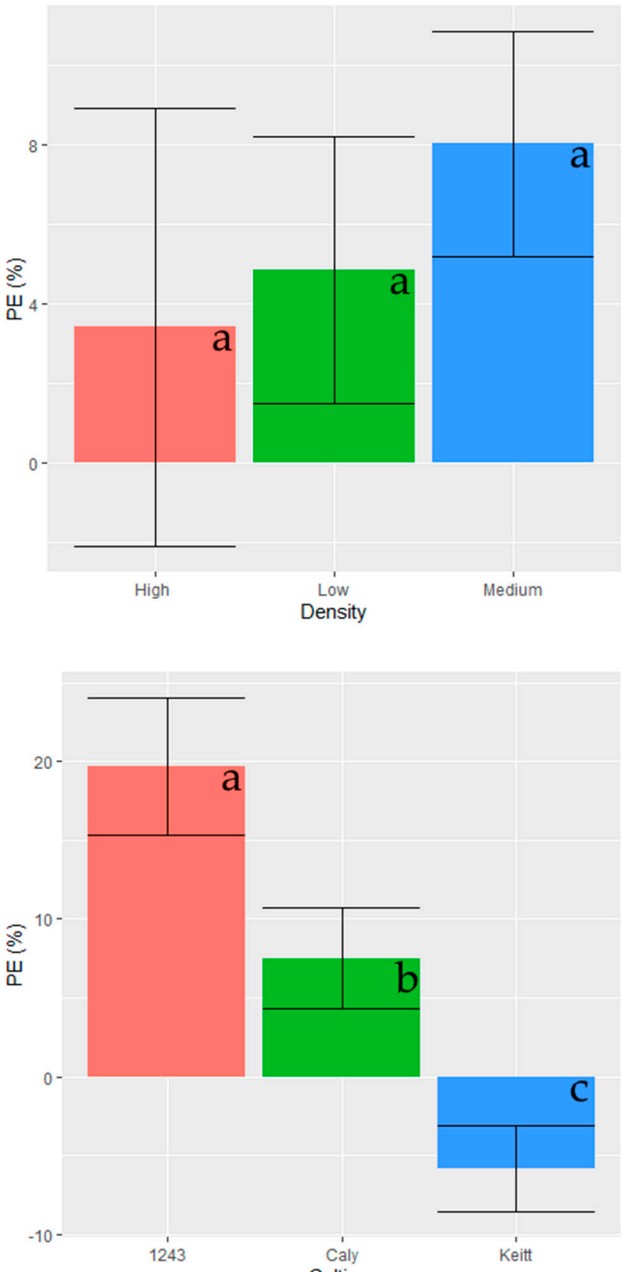

**Figure 3.** Average percentage error of multi-view count of fruit per tree relative to harvest count, presented by training system for all cultivars and densities (top panel); by tree density, for all cultivars and training systems except trellis (middle panel); and by cultivar, for all densities and training systems except trellis (bottom panel). Bars represent ± SD, significant differences denoted with letters a, b and c to $p < 0.05$. Associated data is presented in Appendix A.

### 3.1.3. Down Sampling for Machine Vision Estimation of Fruit Load

A useful discussion of sampling requirements for fruit load estimation is provided by Koirala et al. [4], Wulfsohn [18]. Given that the required number of trees for an accurate estimate of orchard fruit load under a simple random sampling strategy was typically <100 trees for a given orchard Gongal et al. [1,12], it appears redundant to acquire machine vision estimates of all trees in each orchard for an estimation of orchard fruit load. Driving fewer rows also reduces the time and labor requirement of this task, increasing practicality for farm implementation. The statistical estimate of required tree number was, however, based on random sampling, while the machine vision method involves driving of complete

rows, i.e., systematic sampling. More frequent spatial sampling also provides the additional benefit of providing information on the spatial distribution of fruit load.

The variation in average number of fruits per tree side was assessed using the multi-view method for rows within each of five orchards (Table 1). A high SD on fruit number per tree was generally associated with a high SD on row averages for a given orchard, except for orchard 31, i.e., variation in fruit load across rows was greater than in the direction of the rows in orchard 31, such that row averages were similar. For orchard 8, the percentage error of the average fruit load of all rows were within ±10% of the orchard average (Table 1). In this case imaging any single row would provide a reasonable estimate of orchard fruit load. High between-row variation in orchard 23 was associated with an increased number of rows with more than 10% of rows above or below the orchard average (Table 1).

Average fruit number per tree side was estimated with down sampling of row data from five orchards. Data was down sampled to use of every second, third or up to every ninth row, for all possible start rows for each sampling frequency. An estimate error that was >±10% of the whole orchard estimate occurred with down-sampling to every 6th row or less in orchard 23 (Table 1).

Imaging of every third row is recommended as a compromise between sample number, task effort and provision of a map showing spatial variability in fruit load. For example, with sampling of every third row, a 50,000-tree farm was imaged in 20 h, i.e., across two days (see Section 3.3).

**Table 1.** Characterization of five orchards varying in location and cultivar in terms of number (#) of rows and tree, mean and SD of machine vision estimated fruit number per tree side, SD of the average fruit count per tree side for individual rows, the number of rows for which the fruit count per tree side was >±10% of the average value of the entire orchard, and the number of iterations of a given down sampling interval that were >±10% of the orchard average value. Detail is also given for the case of sampling every 6th row, for all start row possibilities. Iterations which exceeded ±10% of the whole orchard estimate are marked by a (*).

| Orchard | 8 | 23 | 28 | 31 | 38 |
|---|---|---|---|---|---|
| Region | NT | NQLD | CQLD | SQLD | SQLD |
| Cultivar | Caly | Keitt | HG | Caly | HG |
| # rows | 18 | 24 | 8 | 20 | 36 |
| # trees | 3474 | 1406 | 2128 | 4650 | 3068 |
| Mean (fruit #/tree side) | 41.8 | 50 | 29.6 | 45.7 | 36.8 |
| SD (fruit #/tree side) | 13.9 | 21.9 | 11.8 | 17.8 | 15.7 |
| SD of row average (fruit #/tree side) | 2.3 | 8.9 | 3.4 | 3.9 | 6.8 |
| # rows > ±10% of orchard fruit #/tree side | 0 | 13 | 3 | 1 | 25 |
| Row sampling interval | # estimates > ±10% of row mean (fruit #/tree side) | | | | |
| every second row | 0 | 0 | 0 | 0 | 0 |
| every third row | 0 | 0 | 0 | 0 | 0 |
| every fourth row | 0 | 0 | 0 | 0 | 0 |
| every fifth row | 0 | 0 | 0 | 0 | 0 |
| every sixth row | 0 | 1 | 0 | 0 | 0 |
| every seventh row | 0 | 0 | 2 | 0 | 0 |
| every eighth row | 0 | 2 | 2 | 0 | 0 |
| every ninth row | 0 | 2 | 3 | 0 | 0 |
| | # fruit/tree side | | | | |
| Sampling every 6th row, given start row: | | | | | |
| 1 | 42.1 | 49.8 | 28.3 | 44.7 | 34.3 |
| 2 | 42.6 | 46.1 | 27.0 | 46.3 | 37.3 |
| 3 | 40.4 | 48.0 | 32.5 | 44.1 | 39.2 |
| 4 | 40.5 | 50.8 | 30.1 | 46.6 | 39.0 |
| 5 | 42.6 | 55.9 * | 32.0 | 47.7 | 37.4 |
| 6 | 42.7 | 52.3 | 28.6 | 45.3 | 33.4 |

### 3.1.4. Crop Timing

The object size limit used in the machine vision method places a constraint on the size of fruit that will be detected and counted. For example, Figure 4 presents images from orchard 12 of a single tree at weekly intervals, with the number of fruits detected increasing as fruit increased in size. This trend is also evident across the whole orchard (Figure 5). For orchard 28 in 2020–2021, multi-view counts appeared to plateau then increased by a further 29% of the final tally. The final count was 106% of the packhouse count (see also Section 3.2.1). Thus, a time-series machine vision assessment of a given orchard allows insight into the spread of crop maturity.

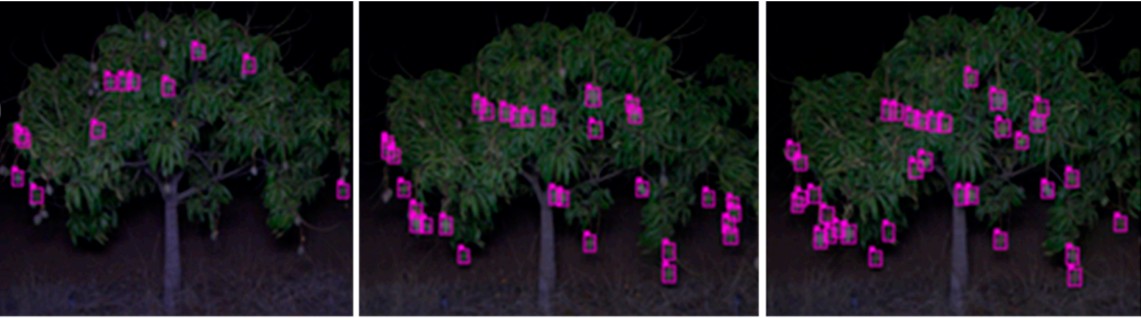

**Figure 4.** Images of a single tree from orchard 12 at weekly intervals, with bounding boxes shown on detected fruit.

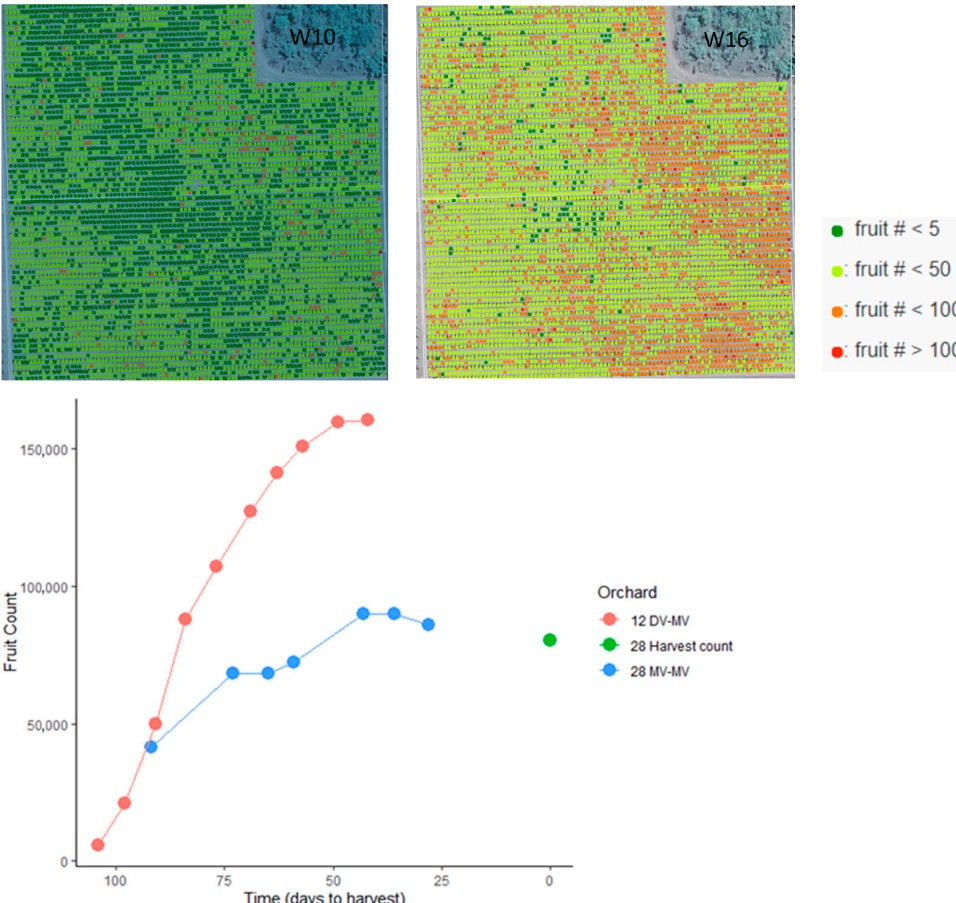

**Figure 5.** Top panel: Fruit count per tree by dual-view machine vision at 10 and 16 weeks (left and right panels, respectively) from flowering for orchard 12 (2018–2019). Each dot represents a tree side, with color indicating fruit number. Bottom panel: Time course of fruit count for the same orchard (orange) and for orchard 28 (2020–2021 season) (blue). Harvest occurred 6 and 4 weeks after the last assessment, respectively. Packhouse count for orchard 28 is shown at time 0.

Repeated machine vision estimates of fruit load are therefore required in seasons with extended flowering events which result in multiple harvest events. Appropriate timing of imaging could allow estimation of the fruit load associated with separate harvest events.

### 3.2. Orchard Estimates

### 3.2.1. Method Comparisons 2019–2020

Four methods of fruit load estimation were compared to the packhouse record across multiple orchards (Table 2). Across the nine orchards for which results for all methods were available, the FARM, CAL, MV-Raw and MV-CAL methods achieved an APE on packhouse counts of 26, 13, 11 and 17%, with SD of 4, 11, 8 and 11%, respectively, with significance differences at a 95% level (Table 2). The MV-raw technique provided the lowest APE, although it was not significantly different to the CAL result (Table 2). For these nine orchards, the $R^2$ of the linear correlation between orchard fruit load estimates and packhouse count was 0.929, 0.988, 0.984 and 0.972 for FARM, CAL, MV-raw and MV-adj estimates. Across all 20 assessed orchards, the MV-Raw technique achieved an APE of 11% (Table 2, Figure 6). These results compare favorably to the few available published estimates of whole orchard yield (e.g., APE of 2 to 84% on yield of 15 mango orchards Sarron et al. [11], 6 and 15% for two kiwifruit orchards [10], 5 to 15% across five mango orchards [4], 0 to 28% for five mango orchards [13] and a MAPE of 18% for 20 apple trees Gongal et al. [1]).

**Table 2.** Packhouse fruit count and the absolute percentage error of fruit load estimates made by (i) farm management as a manual count of a systematic sample of trees (FARM), (ii) a manual count of calibration trees multiplied by tree number (CAL), (iii) the raw machine vision count (MV-Raw) and (iv) a machine vision count adjusted by an 'occlusion' factor calculated of the calibration trees (MV-CAL). Calibration trees were located in the orchards highlighted by bold italic font for a given management zone. Occlusion factors calculated from these trees were applied to other orchards of the same management zone for calculation of CAL and MV-CAL. In orchards designated by *, packhouse fruit numbers represents estimates based on tray numbers and tray sizes. Significant differences ($p > 0.05$) between treatment averages are denoted with letters.

| 2019–2020 | | | | | | |
|---|---|---|---|---|---|---|
| **Zone** | **Orchard** | **Packhouse (#Fruit)** | **FARM (%)** | **CAL (%)** | **MV-Raw (%)** | **MV-CAL (%)** |
| 1 | 1 * | 188,296 | 35 | 20 | | |
| *1* | *2 *| *173,303* | 30 | 9 | | |
| 1 | 3 * | 83,660 | 23 | 4 | | |
| 1 | 4 * | 52,651 | 8 | 23 | | |
| 2 | 5 * | 277,982 | 26 | 16 | | |
| *2* | *6 *| *308,838* | 24 | 16 | 6 | 8 |
| 2 | 7 * | 162,579 | 57 | 19 | | |
| *3* | *8 *| *172,059* | 14 | 28 | 16 | 1 |
| 3 | 9 * | 640,277 | 8 | 4 | | |
| 4 | 10 * | 242,197 | 14 | 6 | 5 | 20 |
| 4 | 11 * | 326,706 | 38 | 1 | 1 | 20 |
| *4* | *12 *| *326,426* | 40 | 10 | 3 | 16 |
| 4 | 13 * | 120,424 | 36 | 12 | 5 | 20 |
| | 14 * | 494,141 | 9 | 7 | 5 | 8 |
| *5* | *15 *| *499,587* | 27 | 12 | 34 | 30 |
| 5 | 16 * | 241,627 | 19 | 18 | | |
| | 18 * | 1,911,894 | 11 | 2 | | |
| | 19 * | 352,368 | 46 | 81 | | |
| | 20 * | 1,640,236 | 29 | 21 | 25 | 29 |
| | 21 * | 919,219 | 39 | 2 | | |
| | 22 * | 43,888 | | 23 | 9 | 26 |

**Table 2.** *Cont.*

**2019–2020**

| Zone | Orchard | Packhouse (#Fruit) | FARM (%) | CAL (%) | MV-Raw (%) | MV-CAL (%) |
|---|---|---|---|---|---|---|
| | 23 * | 71,596 | | 42 | 2 | 5 |
| | 24 | 190,966 | | 28 | | |
| | 25 | 60,416 | | 17 | | |
| | 26 | 87,750 | | 28 | | |
| | 27 | 252,870 | | 1 | | |
| | 28 * | 68,572 | | 5 | 19 | 5 |
| | 30 * | 97,480 | | | 8 | 53 |
| | 31 | 194,511 | | 26 | 9 | 13 |
| | 32 | 149,145 | | 10 | | |
| | 33 | 151,740 | | 9 | 18 | 6 |
| | 34 | 233,546 | | 26 | | |
| | 35 * | 84,635 | | 10 | 9 | 15 |
| | 36 * | 101,766 | | 34 | 10 | 15 |
| | 37 * | 264,801 | | 32 | 10 | 14 |
| | 38 * | 104,642 | | 26 | 8 | 8 |
| | Walkamin | 87,240 | | | 8 | |
| | AVG | | 26.7 | 17.9 | 10.5 | 16.4 |
| | STD | | 13.7 | 15.2 | 8.2 | 12.1 |
| | AVG (9 orchards) | | 25.7a | 12.6ab | 11.1b | 16.9ab |
| | STD (9 orchards) | | 11.3 | 8.2 | 11.4 | 9.7 |

**2020–2021**

| Zone | Orchard | Packhouse (#Fruit) | FARM (%) | CAL (%) | MV-Raw (%) | MV-CAL (%) |
|---|---|---|---|---|---|---|
| | 1–16 * | 3,039,052 | 35 | 17 | 32 | 9 |
| | 18 * | 1,814,684 | 23 | 22 | 20 | 23 |
| | 19 * | 1,533,868 | 14 | 10 | 24 | 19 |
| | 20 * | 1,263,408 | 22 | 25 | 36 | 23 |
| | 21 * | 1,206,123 | 35 | 24 | 26 | 13 |
| | 24 | 23,807 | 41 | | 41 | 76 |
| | 28 * | 80,333 | | | 7 | |
| | 30.1 * | 76,531 | | | 12 | |
| | 30.2 * | 59,461 | | | 19 | |
| | 31 | 186,234 | | | 2 | |
| | 32 | 142,798 | | 31 | 1 | |
| | 33 | 223,608 | | | 8 | |
| | 34 | 223,608 | | | 17 | |
| | 35–36 | 191,902 | | 46 | 27 | |
| | 37 | 91,998 | | 41 | 5 | |
| | 38 | 89,389 | | 27 | 10 | |
| | 40–41 | 589,800 | | 2 | 7 | 4 |
| | 42 | 2,052,195 | 17 | | 17 | |
| | 43 | 199,656 | 28 | 3 | 3 | 6 |
| | AVG | | 27 | 23 | 17 | 22 |
| | STD | | 10 | 14 | 12 | 23 |
| | AVG (6 orchards) | | 26.2a | 16.8a | 23.5a | 15.5a |
| | STD (6 orchards) | | 8.2 | 8.8 | 11.6 | 7.3 |

The use of an occlusion correction factor calculated from the human count of the calibration trees did not improve the orchard load estimate in comparison to the packhouse count in 16 of 20 orchards in the 2019–2020 season, and MV-CAL estimates demonstrated higher variability than the MV-Raw estimates (Table 2). This result indicates that the correction factor for proportion of occluded fruit derived from count of the CAL trees was not representative of the whole orchard. Use of the multi-view machine vision estimate without correction is recommended, with the imaging method restricted to use in orchards with open canopies.

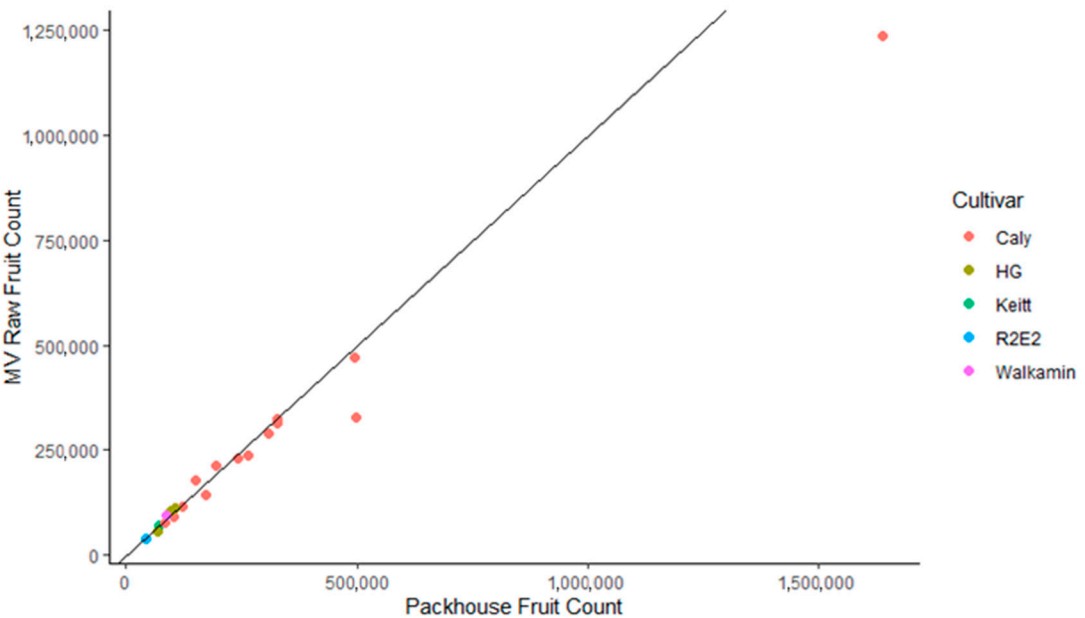

**Figure 6.** Scatter plot of raw multi-view machine vision estimates of 2019–2020 orchard fruit load against packhouse records, $n$ = 20. Data is fitted by a linear regression model, $R^2$ = 0.98, slope = 0.74 and $p < 0.001$.

There remains a need to develop better tools for estimation of an occlusion factor for a given orchard. Koirala et al. [16] unsuccessfully attempted estimation of a per tree occlusion factor based on machine vision features such as the proportion of partly occluded to non-occluded fruit. A random forest model on the ratio of machine vision to packhouse count using inputs of various orchard attributes explained only 39% of the variance in the ratio, with 10, 8, 6, 4 and 3% attributed to tree age, SD of tree crown area, mean of tree crown area row spacing and tree spacing, respectively (Appendix B). Tree age is loosely related to canopy size and density, and a high SD on tree crown area is likely to be associated with higher variation in canopy size and density, and thus in the occluded fruit ratio. The explained variance was, however, still too low to be of practical use.

The CAL manual count method achieved a reasonable estimate of orchard fruit load in many cases (Table 2). In some cases, poor results were attributable to a poor relationship between human count of fruit on tree and harvest count, e.g., for orchard 36 the FARM APE was 34% and the percentage difference between on-tree count and per tree harvest count of 18 trees was also 34% (data not shown). Errors in human count of fruit per tree will propagate through to orchard fruit load estimates for any methods that uses human counts as an input.

The FARM method under-performed relative to the CAL method, despite the count of more trees. This result is ascribed to the sampling strategy used in the FARM method, a pseudo systematic approach involving a single transect across each orchard. In comparison, the CAL method involved random sampling within a stratified sampling approach involving classification to three classes on NDVI values.

### 3.2.2. Method Comparisons 2020–2021

The multi-view machine vision results were under-estimates of packhouse count for several orchards in the 2020–2021 season. Based on farm observations, this result was ascribed to an extended flowering period such that a single machine vision assessment event occurring at the time of stone hardening of fruit from the main flowering event failed to count later maturing fruit (see Section 3.1.4), rather than to a change in the canopy occlusion factor between seasons. The relationship between the multi-view count and the packhouse count (Figure 7) was characterized by a high $R^2$ but a slope less than unity, indicates that the proportion of crop that was late was similar in all orchards of this farm.

A similar result was obtained on another farm of the same cultivar (slope of 0.77, $R^2 = 0.97$, data not shown).

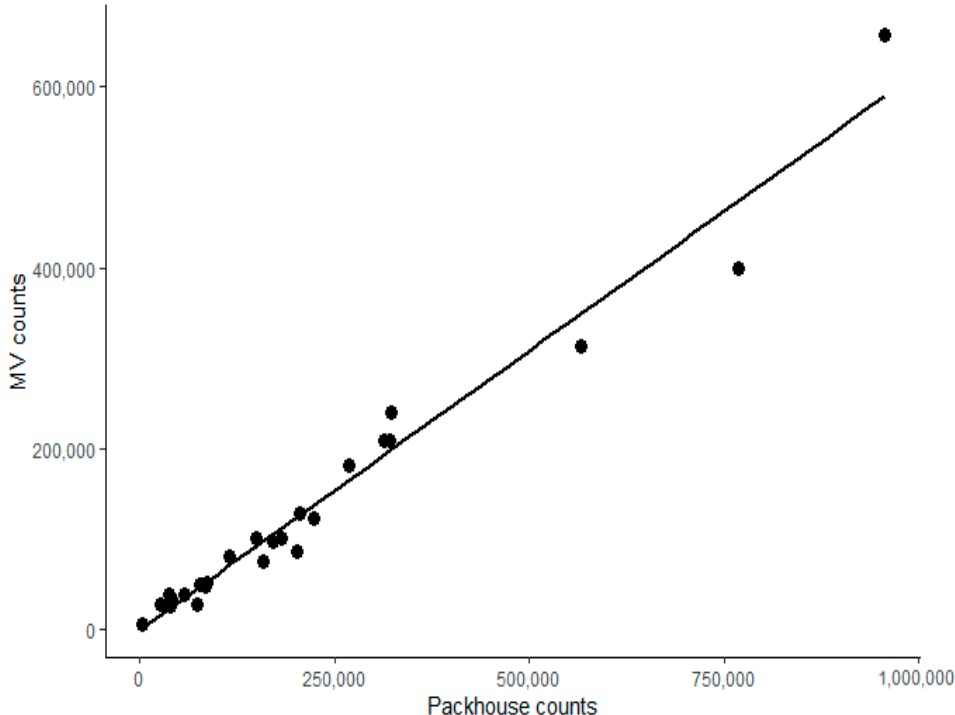

**Figure 7.** 2020–2021 season multi-view machine vision count of fruit per orchard fruit plotted against packhouse fruit count for 20 orchards from one farm. Data is fitted by a linear regression model, $R^2 = 0.97$, slope = 0.62, $p < 0.001$.

The APE of MV-Raw, MV-CAL, CAL and FARM methods across all orchards measured in the 2020–2021 season was 17, 23, 22 and 27%, respectively (Table 2). For the six orchards for which all methods were available there was no significant difference at a 95% confidence level between methods. Manual count-based estimates (CAL, MV-CAL) exceeded the machine vision estimate for orchards 1–16 as the manual count was made three weeks after the machine vision estimate. Due to a prolonged flowering period, additional fruit reached stone hardening stage in this period. In this case, the average percentage error of the multi-view count was improved by use of a correction factor (PE of 32 and 9% for MV-Raw and MV-CAL methods, respectively) (Table 2).

*3.3. Use Cases*

The machine vision method for estimation of tree crop load allows for assessment of all trees in each orchard, at a drive time of approximately 21 min for a 3 ha, 1000 tree orchard, compared to approximately 3 h for the manual count 18 trees. Data processing time was equivalent to imaging time, allowing for same day provision of fruit load estimates to inform farm planning

Machine vision fruit load estimates can also be used for purposes beyond an orchard fruit load estimate. The multi-view method was based on continuous imagery of the row at 10 fps, with an estimate of fruit load per tree achieved from cumulative fruit counts taken at the tree spacing interval, commonly 3 m. This data approximates a per tree estimate. Three use cases of this data are documented below:

(i)    Per tree fruit load can used to generate a frequency distribution of tree fruit load. For example, Figure 8 displays the frequency distribution for a single orchard in two seasons. The agronomic use of such information remains to be explored.

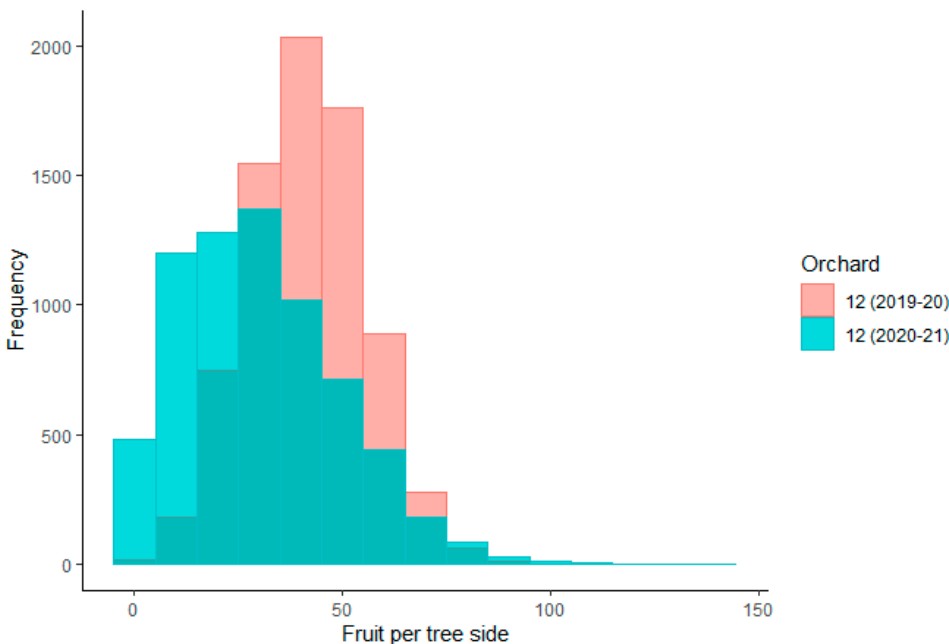

**Figure 8.** Frequency distribution for tree fruit load for tree side of orchard 12 in 2019–2020 and 2020–2021 season (11-year-old trees in 2020–2021 season).

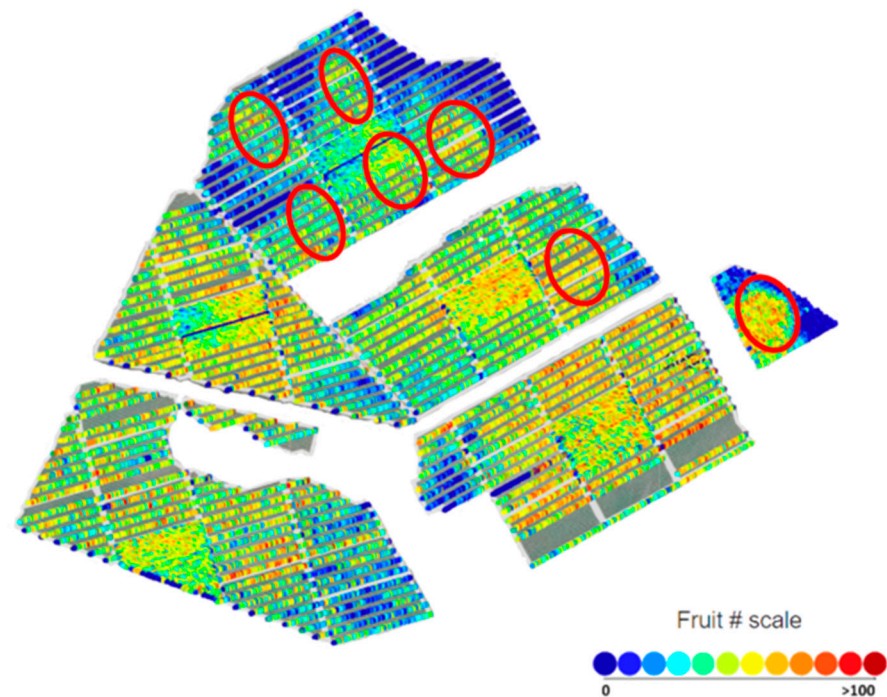

**Figure 9.** *Cont.*

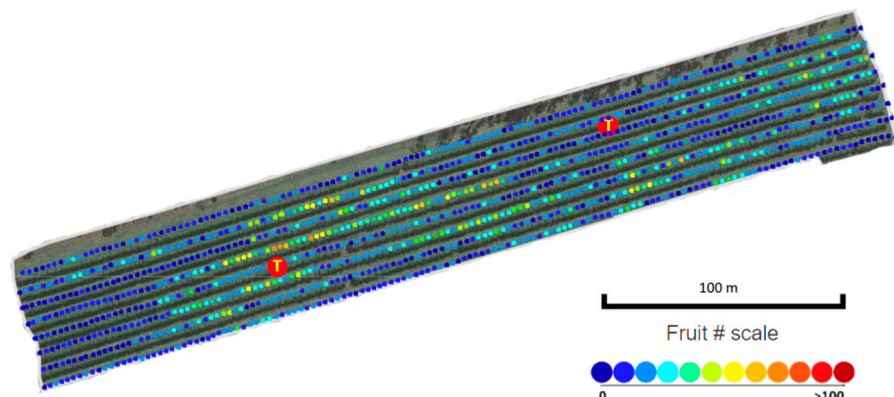

**Figure 9.** Fruit load 'heat map' for a farm affected by frost at flowering (top panel), and of early fruit in an orchard (bottom panel). Red 'T' denotes location of temperature sensors. Red ovals indicate area of influence of frost fan (top panel).

(ii)   A count increment at the tree spacing interval allows for display of a 'heat map' of fruit load across a farm. In one farm this display was interpretable in terms of the location and speed of operation of fans that had protected flowers on the farm during a frost event, with the display used to guide the subsequent placement of additional fans (Figure 9, top panel).

(iii)  An imaging event early in the fruit development period captured the location of early maturing fruit, associated with early flowering trees (Figure 9, bottom panel). This information was used by farm management to guide a selective early harvest event, responding to market demand.

## 4. Conclusions

The use of the multi-view machine vision method is recommended for mango fruit load estimation, with restriction to use with orchards with trees with open canopies that afford view of fruit from interrow positions. For example, reasonable estimates were achieved on hedge, single leader and conventional systems across three cultivars and three planting densities (APE of 3.4, 5.0 and 8.2%, respectively). The APE of multi-view estimates of mango fruit load was generally lower 'as-is' than when corrected for the proportion of occluded fruit as estimated from a sample of orchard trees. This result indicates that the trees sampled for estimation of the occlusion factor did not adequately represent the orchard. The need for repeat estimations during a season with an extended flowering period was recognized, and down-sampling to imaging of every third inter-row was recommended as a compromise between required effort and acquisition of information on spatial variation in fruit load across the orchard. Additional value can be created by imaging of whole orchards in terms of the spatial information on fruit load levels across orchards, e.g., for selection of areas for early harvesting and for mapping of damaged areas.

Several sources of error for the packhouse record of harvested fruit number were documented. It is recommended that researchers probe the accuracy of these values if used as the reference value against which in-field fruit load estimates are compared.

**Author Contributions:** Conceptualization, N.T.A., K.B.W., A.K., Z.W. and M.H.A.; Methodology, N.T.A., K.B.W., A.K. and Z.W.; Software, N.T.A., A.K. and Z.W.; Validation, N.T.A., K.B.W., A.K., Z.W., G.R.D. and M.H.A.; Formal Analysis, N.T.A., K.B.W. and P.S.; Investigation, N.T.A., K.B.W., A.K., Z.W., M.H.A. and G.R.D.; Resources, K.B.W. and A.J.R.; Data Curation, N.T.A. and K.B.W.; Writing—Original Draft Preparation, N.T.A. and K.B.W.; Writing—Review & Editing, N.T.A., K.B.W., G.R.D., P.S. and A.J.R.; Visualization, N.T.A. and Z.W.; Supervision, K.B.W. and A.J.R.; Project Administration, K.B.W. and A.J.R.; Funding Acquisition, K.B.W. and A.J.R. All authors have read and agreed to the published version of the manuscript.

**Funding:** This research was funded by the Australian Government Department of Agriculture and Water Resources as part of its Rural R&D for Profit program through Hort Innovation, with support from Central Queensland University, project ST19009. The Walkamin planting systems trial was established via the 'Transforming subtropical and tropical tree productivity' project, a collaboration between Hort Innovation using the across industry R&D levy, and co-investment from DAF, Queensland Alliance for Agriculture and Food Innovation and the Australian government.

**Acknowledgments:** We acknowledge the support of Alan Niscioli and Danilo Guinto of the Northern Territory's Department of Industry, Tourism and Trade and Dale Bennet and Ebony Faichney of Queensland's Department of Agriculture and Fisheries (DAF) for manual field counts of tree fruit load, Sarah Hain of the Australian Mango Industry Association and the input of the many participating farms.

**Conflicts of Interest:** The authors declare no conflict of interest.

## Appendix A

Fruit count per tree by multi view machine vision expressed as a ratio of harvest count per tree (average and SD) for tree rows, in a factorial combination of three densities of planting, four management canopy systems and three cultivars.

**Table A1.** 2019–2020 orchard characteristics. Cultivar abbreviations are Caly for Calypso®, HG for Honey Gold and region abbreviations are NT for Northern Territory, NQLD for north Queensland, CQLD for central Queensland and SQLD for south Queensland. '-' represents data not available. (Orchard numbering is tied to other tables).

| Density | System | Cultivar | MV/Harvest | SD | n (Rows) |
|---------|--------|----------|------------|------|----------|
| High | Hedge | 1243 | 1.32 | 0.22 | 8 |
| High | Hedge | Caly | 0.85 | 0.18 | 8 |
| High | Hedge | Keitt | 0.96 | 0.18 | 8 |
| High | Trellis | 1243 | 1.47 | 0.18 | 8 |
| High | Trellis | Caly | 1.16 | 0.11 | 9 |
| High | Trellis | Keitt | 1.21 | 0.29 | 10 |
| Medium | Single Leader | 1243 | 1.22 | 0.16 | 6 |
| Medium | Single Leader | Caly | 1.13 | 0.14 | 10 |
| Medium | Single Leader | Keitt | 0.87 | 0.15 | 10 |
| Medium | Conventional | 1243 | 1.22 | 0.23 | 7 |
| Medium | Conventional | Caly | 1.14 | 0.18 | 10 |
| Medium | Conventional | Keitt | 0.98 | 0.13 | 8 |
| Low | Conventional | 1243 | 0.99 | 0.17 | 6 |
| Low | Conventional | Caly | 1.15 | 0.14 | 9 |
| Low | Conventional | Keitt | 0.98 | 0.10 | 7 |

## Appendix B

*Appendix B.1. Variable Importance for the Ratio of Multi-View to Packhouse Count*

Appendix B.1.1. Method

A Geographic Object-Based Image Analysis (GEOBIA) approach was used for tree crown delineation of individual mango trees. The method used spectral reflectance variability as an attribute for discriminating features along with textural and contextual information to delineate mango trees [19,20]. PAN sharpened WorldView-3 images of 0.31 m spatial resolution was used in GEOBIA to delineate tree crowns using eCognistion Developer 8 software [21]. Tree Crown Area (TCA) statistics were then estimated for individual trees in each orchard. TCA was utilized in analysis of variable importance of orchard attributes in relation to the accuracy of fruit load estimates by machine vision, relative to packhouse fruit counts. This analysis was undertaken for a subset of orchards for which satellite imagery, machine vision data and orchard descriptors of cultivar, tree and row spacing and tree age were available (Table A2).

**Table A2.** 2019–2020 orchard characteristics. Cultivar abbreviations are Caly for Calypso®, HG for Honey Gold and region abbreviations are NT for Northern Territory, NQLD for north Queensland, CQLD for central Queensland and SQLD for south Queensland. '-' represents data not available. Orchard numbering is tied to other tables and is an extension of that used in [12].

| Orchard # | Region | Cultivar | Tree Spacing (m) | Inter Row Spacing (m) | Tree Planting Date | Average Tree Crown Area (m²) | SD Tree Crown Area (m²) |
|---|---|---|---|---|---|---|---|
| 6 | NT | Caly | 3 | 8 | 2010 | 8.65 | 2.73 |
| 8 | NT | Caly | 3 | 8 | 2008 | 7.76 | 2.46 |
| 10 | NT | Caly | 3 | 8 | 2010 | 8.64 | 2.31 |
| 11 | NT | Caly | 3 | 8 | 2010 | 7.04 | 2.35 |
| 12 | NT | Caly | 3 | 8 | 2010 | 7.55 | 2.70 |
| 13 | NT | Caly | 3 | 8 | 2010 | 5.99 | 2.16 |
| 14 | NT | Caly | 4 | 9 | 2003 | 15.88 | 4.99 |
| 15 | NT | Caly | 3 | 8 | 2000 | 15.40 | 4.28 |
| 20 | NQLD | Caly | 4.5 | 8 | 2007 | 13.48 | 4.12 |
| 22 | NQLD | R2E2 | 5 | 8 | 2001 | 18.35 | 5.90 |
| 23 | NQLD | Keitt | 5 | 7.5 | 1996 | 9.46 | 3.46 |
| 28 | CQLD | HG | 3 | 7 | 2001 | - | - |
| 30 | CQLD | HG | 3.75 | 7 | 2013 | - | - |
| 31 | SQLD | Caly | 3.5 | 9.5 | 2014 | 3.94 | 2.13 |
| 33 | SQLD | Caly | 3.5 | 9.5 | 2014 | 4.98 | 3.16 |
| 35 | SQLD | Caly | 4 | 9 | 2004 | - | - |
| 36 | SQLD | Caly | 4 | 9 | 2004 | - | - |
| 37 | SQLD | Caly | 4 | 9 | 2004 | 19.77 | 6.87 |
| 38 | SQLD | HG | 4 | 8 | 2012 | 13.15 | 3.00 |
| Walkamin | NQLD | multiple | multiple | multiple | 2013 | - | - |

Appendix B.1.2. Results

With the multi view machine vision method restricted to use with relatively open canopies, it is important to be able to anticipate when an orchard is unsuited for the method. To this end a Random Forest model was developed using 500 decision trees for the ratio MV to packhouse fruit counts based on orchard attributes of tree age, SD of TCA, mean TCA, row spacing and tree spacing for 15 orchards (from Table A2).

The model explained only 39% of the variance in the ratio, in terms of cross-validation results ($R^2_{cv}$ = 0.39, RMSECV = 0.09), with the most important predictor being tree age (Figure 1). This result is consistent with the observations that there is risk of double counting of fruit from the two sides of the tree in young, small trees, while old, large trees can have dense canopies, with risk of increased fruit occlusion. However, this relationship can be altered by canopy management practices. It was hypothesized that trees with large crown areas would also be associated with higher levels of occluded fruit, and thus a decreased machine vision to harvest ratio. In practice, however, the standard deviation of TCA was more important than the mean. Inter-row and tree spacing contributed relatively little to the explained variance.

There remains a need to develop better tools for estimation of an occlusion factor for a given orchard.

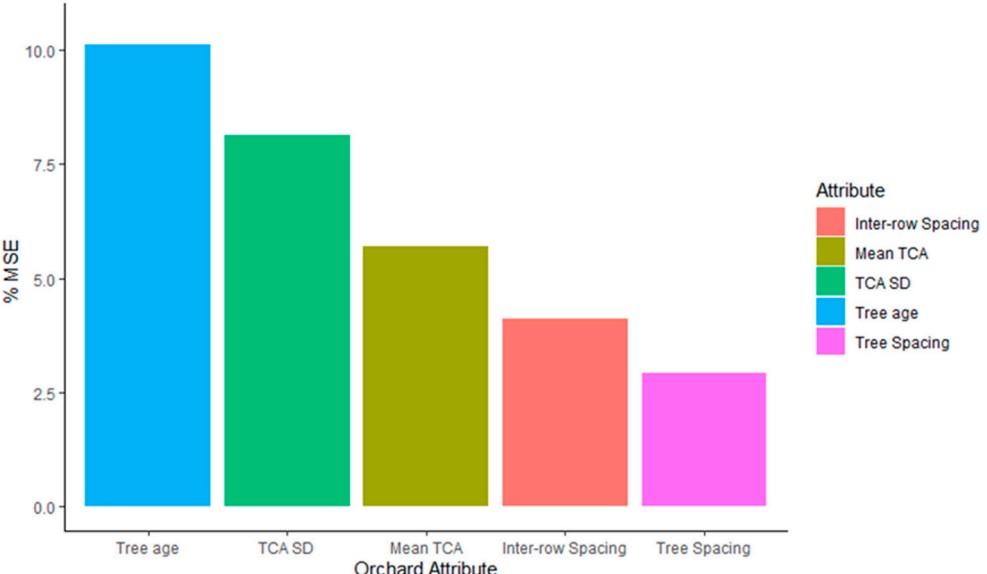

**Figure A1.** Percentage of mean square error (MSE) attributed to five orchard attributes (tree age, SD and mean of tree crown area, inter-row and tree spacing) for the prediction of the ratio of multi view machine vision (MV) to packhouse counts using a Random Forest model.

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
