# Peer review of "Estimation of Fruit Load in Australian Mango Orchards Using Machine Vision"

_agronomy, doi:10.3390/agronomy11091711_

Round 1
Reviewer 1 Report
The manuscript has improved in many aspects, notably in the adoption of consistent error metrics that support the aim of the research.
However, the length and complexity of the manuscript are still not justified given the scope of the research. Estimation of manual count is ancillary to the aim of evaluating machine vision accuracy, yet occupies several pages of the text. Similarly, comparison of single vs. multi-view machine vision is evaluated, but ultimately the multi-view is selected as the method of choice for the majority of experiment. I strongly suggest a re-evaluation of what results are considered critical to the aim of the research, and place material not fitting that criteria in a supplementary section. These actions I believe it would greatly improve the length and readability of the manuscript.
Author Response
The manuscript has improved in many aspects, notably in the adoption of consistent error metrics that support the aim of the research.
>>Thank you
However, the length and complexity of the manuscript are still not justified given the scope of the research. Estimation of manual count is ancillary to the aim of evaluating machine vision accuracy, yet occupies several pages of the text. Similarly, comparison of single vs. multi-view machine vision is evaluated, but ultimately the multi-view is selected as the method of choice for the majority of experiment. I strongly suggest a re-evaluation of what results are considered critical to the aim of the research, and place material not fitting that criteria in a supplementary section. These actions I believe it would greatly improve the length and readability of the manuscript.
>> We have focused the manuscript to an evaluation of machine vision estimates, and so markedly shortened it.
Reviewer 2 Report
The manuscript contains very interesting and useful data. It is well written. However, the manuscript needs revision.
- The section Introduction includes too much duplicate information in relation to the previous publication of the authors: [1] Anderson, N., Underwood, J., Rahman, M., Robson, A., & Walsh, K. (2019). Estimation of fruit load in mango orchards: tree sampling considerations and use of machine vision and satellite imagery. Precision Agriculture, 20(4), 823-839.
- lines 48-57 and 58-63: The citations are missing. Is it unpublished information?
- Originality or novelty and the aim of the paper should be more emphasized in the last paragraph of the Introduction.
- line 182: "tree age and orchard layout" - that should be detailed here
- lines 201-205: Why is the number of repetitions not the same in all cases?
- Subsection 2.3. Machine vision system (lines 230-254): Is it possible to add a schema or a photo of the machine vision system?
- Section Discussion does not include enough references. Anderson et al. [1] is mainly cited.
Author Response
The manuscript contains very interesting and useful data. It is well written.
>>Thank you
However, the manuscript needs revision.
- The section Introduction includes too much duplicate information in relation to the previous publication of the authors: [1] Anderson, N., Underwood, J., Rahman, M., Robson, A., & Walsh, K. (2019). Estimation of fruit load in mango orchards: tree sampling considerations and use of machine vision and satellite imagery. Precision Agriculture, 20(4), 823-839.
>>The manual sampling section has been removed and the Introduction also tightened in other respects.
- lines 48-57 and 58-63: The citations are missing. Is it unpublished information?
>>This represents our knowledge of these commercial systems. Text has been altered.
- Originality or novelty and the aim of the paper should be more emphasized in the last paragraph of the Introduction.
>>Text modified.
- line 182: "tree age and orchard layout" - that should be detailed here <<This information was in Table 2. This table is now in Appendix B, but better reference is made to it in text.
- lines 201-205: Why is the number of repetitions not the same in all cases?
<<This was because of the design of the high density experimental orchard, which has variation in tree density but same row lengths, and thus variation in tree numbers per treatment. Comment has been added to text.
- Subsection 2.3. Machine vision system (lines 230-254): Is it possible to add a schema or a photo of the machine vision system? >>Added
- Section Discussion does not include enough references. Anderson et al. [1] is mainly cited. >>Comparison to several studies, were relevant, have been added
Reviewer 3 Report
The present manuscript presents data on pre-harvest assessment of mango yields. Assessing expected harvest quantities is important for packinghouses and marketing agents for their preparation before the season. The authors present a very large set of results, comparing different methods for assessing fruit numbers. They compare manual counts to various aspects of machine vision – using either dual-methods (acquiring images from two sides of the trees) or multi-view methodology (taking multiple images from different angels and applying computer algorithms to count each fruit only once).
The paper is very long (28 publisher-ready manuscript pages), and very technical. I also found it hard to read and repetitive. I think that it should be shortened considerately if accepted for publication. The topic of the paper seems more applicable to a journal on precision agriculture or agricultural engineering then to a journal focused on agronomy and horticultural practices.
Most of the methods described here were already developed and published (see the very recent papers written by the authors in the reference list -1, 3, 12, 16, 19 as well as others). The scope of the paper is mainly in comparing the methods on a very large scale. However, as the main methods were described before, I found only limited novelty in the current results.
Several additional specific comments:
Lines 209-201: Manual counts and measurements were performed approximately 6 weeks before harvest. Although this is after the main fruit drop, mango fruits usually drop up to the harvesting date.
Line 803 : reference 3: "Soon to be published" is this manuscript fully accepted? Can it be presented as "in press"? Otherwise, it should be omitted from the reference list.
Figure 9: Please explain why you used a log-log scale (and not regular correlations as presented in Fig 10). Please provide details of the correlation's equation, r2 and p values. Does the double log values reduce the sensitivity of the correlation?
It is suggested that fruit size and age of the fruit are major factors in detection by both manual and automatic counts (Figures 5 and 6). Therefore, it would have been better if a specific time before harvest was used for all measurements in all orchards. In such a large experiments, differences in the time of counting within orchard and between orchards are expected. However, they make the analysis complicated. This can be clearly understood from the authors comment in lines 684-685 where the manual count was performed 22 days after the automatic one. For better comparison of the results, the data in Table 2 and elsewhere should include specific information on the specific dates the different counts were made, using day before harvests as the standard. This will enable a better comparison between different methods, between orchards and between seasons.
Figure 10: pleas not that there are three different figures marked as Figure 10.
Fig. 10 (the last one) – while the use of frequency distribution between orchards can be a tool for understanding physiological issues such as flowering/fruit set/natural drops act., the authors decided to present a rather trivial case, comparing orchard with very young trees to an old orchard. In such a case, the differences are trivial, and the image should be removed. On the contrary, the results in Fig. 11 that present similar data as heatmaps, suggest an important physiological effect for the use of fans to enable fruit set. On the other hand, This is not the main topic of the paper. I think it would have have been better, it the paper was focusing on a specific subject (comparison of different methods and their accuracies) and that the use of it for other, horticultural/physiological aspects of the count data that are only mentioned in this paper, will be presented in a specific manuscript.
I therefore suggest that major changes in the paper, including extensive shortening are required before it could be considered for publicashing in Agronomy.
Author Response
The present manuscript presents data on pre-harvest assessment of mango yields. Assessing expected harvest quantities is important for packinghouses and marketing agents for their preparation before the season. The authors present a very large set of results, comparing different methods for assessing fruit numbers. They compare manual counts to various aspects of machine vision – using either dual-methods (acquiring images from two sides of the trees) or multi-view methodology (taking multiple images from different angels and applying computer algorithms to count each fruit only once).
The paper is very long (28 publisher-ready manuscript pages), and very technical. I also found it hard to read and repetitive. I think that it should be shortened considerately if accepted for publication.
<< The revised manuscript has been focused in its scope, and thus considerably shortened (to 19 pages).
The topic of the paper seems more applicable to a journal on precision agriculture or agricultural engineering then to a journal focused on agronomy and horticultural practices. Most of the methods described here were already developed and published (see the very recent papers written by the authors in the reference list -1, 3, 12, 16, 19 as well as others). The scope of the paper is mainly in comparing the methods on a very large scale. However, as the main methods were described before, I found only limited novelty in the current results.
<<Emphasis is placed in the aim statement at end of the Introduction that this report is not a method development exercise (e.g., on development of a deep learning architecture for use in fruit detection) but a field (or agronomic) evaluation of a system based on machine vision to estimate tree fruit load. There is a big step between achieving 90% detection of fruit in an image or 90% detection of fruit on a single tree to an orchard level estimate at 90% of the actual harvest result.
Several additional specific comments:
Lines 209-201: Manual counts and measurements were performed approximately 6 weeks before harvest. Although this is after the main fruit drop, mango fruits usually drop up to the harvesting date.
<<In our experience in the Australian mango industry, across many orchards and seasons, is that fruit drop after stone-hardening is minimal, except in exception circumstances (irrigation failure etc).
Line 803 : reference 3: "Soon to be published" is this manuscript fully accepted? Can it be presented as "in press"? Otherwise, it should be omitted from the reference list. <<Now published, full citation added. The review paper frames the current experimental paper.
Figure 9: Please explain why you used a log-log scale (and not regular correlations as presented in Fig 10). Please provide details of the correlation's equation, r2 and p values. Does the double log values reduce the sensitivity of the correlation? <<It was a display problem, given the huge range of values. Correlation statistics have been added for the log and linear plots.
It is suggested that fruit size and age of the fruit are major factors in detection by both manual and automatic counts (Figures 5 and 6). Therefore, it would have been better if a specific time before harvest was used for all measurements in all orchards. In such a large experiments, differences in the time of counting within orchard and between orchards are expected. However, they make the analysis complicated. This can be clearly understood from the authors comment in lines 684-685 where the manual count was performed 22 days after the automatic one. For better comparison of the results, the data in Table 2 and elsewhere should include specific information on the specific dates the different counts were made, using day before harvests as the standard. This will enable a better comparison between different methods, between orchards and between seasons. << Targeted discussion of this issue added.
Figure 10: pleas not that there are three different figures marked as Figure 10. << Good grief! Corrected
Fig. 10 (the last one) – while the use of frequency distribution between orchards can be a tool for understanding physiological issues such as flowering/fruit set/natural drops act., the authors decided to present a rather trivial case, comparing orchard with very young trees to an old orchard. In such a case, the differences are trivial, and the image should be removed. >>The figure has been altered to display data of the same orchard in two seasons
On the contrary, the results in Fig. 11 that present similar data as heatmaps, suggest an important physiological effect for the use of fans to enable fruit set. On the other hand, This is not the main topic of the paper. I think it would have have been better, it the paper was focusing on a specific subject (comparison of different methods and their accuracies) and that the use of it for other, horticultural/physiological aspects of the count data that are only mentioned in this paper, will be presented in a specific manuscript. <<With the refocussed and very much shortened manuscript, we believe that these ‘use cases’ are a useful addition, pointing to agronomic uses for yield maps beyond a simple tally of fruit per orchard.
I therefore suggest that major changes in the paper, including extensive shortening are required before it could be considered for publicashing in Agronomy. << Manuscript has been extensively shortened.
Round 2
Reviewer 2 Report
I recommend to accept manuscript in its present form
Author Response
I recommend to accept manuscript in its present form
>> Thank you
Reviewer 3 Report
The authors have moved substantial information into a supplementary file, shortening the manuscript. However, I still find it extensively long. The supplementary material should be limited to less important data (figures and tables), or to some technical methods and these need to be clearly mentioned and referred to in the main file of the manuscript (specifically that they are presented in the online supplementary file – as specific Suppl. Figs. Number or Suppl. Table number). Rather than moving specific tables and figures to the online appendices, the authors moved complete large sections to the appendices without specifically referring to each section in the main text file. Moreover, the authors deleted introductory information important to understand the context of the text in the parts that remain in the main document (for example: in Methods the different orchard description can be in the appendix but it should be referred to in the main text. The Results section can not really start by "In a test of method repeatability two rows of orchard 30… "without any preliminary explanation of the system and orchards. It seems that the authors moved parts to the appendices but did not bother to turn the two document into a one logical piece.
Fig. 1 – I do not think that this figure adds to the paper. This is especially true since the paper is still long. If the editor decides to leave the figure, I suggest (a) acquiring a better picture (b) and clearly indicate the different modules on the figures.
Fig.2 the resolution of the bounding frames in the photos are not clear to detect their colors. Please correct.
Fig 6 (old fig 9). I still do not see why a log-log correlation is performed. It is true that datasets are with wide variation in values. However, I still think that performing the correlation on log – log values reduce the sensitivity of the correlation. I* still suggest to remove this image or to present it with linear axe
Author Response
Reviewer 3:
The authors have moved substantial information into a supplementary file, shortening the manuscript. However, I still find it extensively long. The supplementary material should be limited to less important data (figures and tables), or to some technical methods and these need to be clearly mentioned and referred to in the main file of the manuscript (specifically that they are presented in the online supplementary file – as specific Suppl. Figs. Number or Suppl. Table number). Rather than moving specific tables and figures to the online appendices, the authors moved complete large sections to the appendices without specifically referring to each section in the main text file. Moreover, the authors deleted introductory information important to understand the context of the text in the parts that remain in the main document (for example: in Methods the different orchard description can be in the appendix but it should be referred to in the main text. The Results section can not really start by "In a test of method repeatability two rows of orchard 30… "without any preliminary explanation of the system and orchards. It seems that the authors moved parts to the appendices but did not bother to turn the two document into a one logical piece.
>> We have already shortened the manuscript considerably, by reduction of 9 pages. We wish not to reduce further, but will remove Appendix B on advice of the Editor. I have amended an error at line 119 which referenced Appendix A rather than Appendix B, this gives several orchard attributes to the reader. We believe we have referenced the Appendices sufficiently (lines 119 and 425) so that the reader has been briefed on the orchards.
Fig. 1 – I do not think that this figure adds to the paper. This is especially true since the paper is still long. If the editor decides to leave the figure, I suggest (a) acquiring a better picture (b) and clearly indicate the different modules on the figures.
>> We have changed this image and added labels to components.
Fig.2 the resolution of the bounding frames in the photos are not clear to detect their colors. Please correct.
>> We have enhanced the bounding box frames in figure.
Fig 6 (old fig 9). I still do not see why a log-log correlation is performed. It is true that datasets are with wide variation in values. However, I still think that performing the correlation on log – log values reduce the sensitivity of the correlation. I* still suggest to remove this image or to present it with linear axe
>> We have added the raw plot and removed the log-log statistics from description, the plot is worse to interpret as a single data point is magnitudes larger than all the other data points, thus use of a log scale. We defer to the Editor to keep the raw plot or revert the changes to the log-log scale plot.
Thank you!
This manuscript is a resubmission of an earlier submission. The following is a list of the peer review reports and author responses from that submission.
Round 1
Reviewer 1 Report
The paper improved a lot, but it is still not consistent, it is not easy to follow and needs more works.
Reviewer 2 Report
Sensing technology is fundamental in precision orchard management. The aim of the paper is to in-field counting of fruit on tree using machine vision. The filed machine vision system and analysis always valuable to satisfy specific requirements of high efficiency and feasible applications.
The paper presents the system design with hardware. Experiments were introduced in details. The processing methods were proposed to count fruit load in different season. In general, problems are not solved clearly and logically, as follows:
1/ The image processing logical is not clearly described. More introduction about the motivations and the estimation pipeline should be involved to explain the reasonable design in the research.
2/ The selected cameras are two of Basler acA2440 RGB. DV and MV imaging methods should be strength to show the focal length, distances, camera calibrations and so on. This is the fundamental for image processing.
3/ Although different seasonal data are involved, the ANOVA should be conducted.
4/ Deep learning method is used. The image data set should be introduced and the training results should be estimated by the IoU, loss value and so on. the structure or the training environment also needs be described.
5/ The results should be proved in different seasons and the conclusion should be strength to show how you solve the challenges in the field mongo counting.
Reviewer 3 Report
Overall, the paper reports many valuable findings from a very large sample, but I believe the scope can be narrowed to advance a more focused narrative (and a paper of more reasonable length). Firstly, the introduction is a very extensive literature review, perhaps to the detriment of the paper. I would encourage authors to be more succinct, and report general trends found in other work as opposed to detailed findings. Secondly, I consider sections 3.3 and 3.4 to be the primary findings of the manuscript. Therefore, sections 3.1 and 3.2 could be either reduced and/or portions placed in a supplementary materials.
Many measurements of error are implemented throughout the manuscript (RMSE, percent error, ratios, R2, etc.), but not consistently applied. For instance, line 339 states "The mean percentage error was 7.8% for the 32 orchards (Table 2, Figure 1)". However, no value of 7.8% is reported in either Table 2 or Figure 1, nor is the language "percent error" used in the captions or labels. RMSE is used, but this is very different than % error, and should be interpreted differently (RMSE is in the same unit of measurement as the observation, % error is not). Being more explicit and consistent with application of these error measurements in suggested, particularly since they have different interpretations.
Similarly, the aim of the paper is clearly to use 10% error as a threshold for evaluating various methods. However, RMSE does not contribute to this aim (10 RMSE does not amount to 10% error). And neither does ratio/"relationship" that is reported in sections 3.3 and 3.4. Notably, percent error is not reported at all in sections 3.3 or 3.4! In this way, the results reported do not meaningfully address the aim of the paper as laid out in the introduction.
And finally, a major issue I detect in the manuscript is that the manual fruit counts and packhouse counts are themselves estimation given user repeatability error, and estimation of fruit # based on weight. The authors acknowledge these issues in lines 178-183, and again on lines 375-382, but do not account for these uncertainties when comparing manual counts to machine vision methods. Therefore, the errors and confidence intervals in machine vision estimation are assuredly poor indicators of actual accuracy of the method. More discussion is needed to acknowledge these limitations.
Specific comments:
Line 106: "The first report...made by [8]" - restructure sentence to include author information (similar to line 146)
Line 137: "Error" is written twice - redundant
Line 153: What factors contribute to inter-annual discrepancies? It may be useful to discuss.
Line 419: What us "vision count to manual count relationship"? A slope? An R2? A ratio? Unclear.
Line 421: Consistent in what way?
Figure 4: In the text, panels are referred to as A/B/C. In the caption, panels are top/middle/bottom. Text should be consistent with the formatting. Further, the legend is redundant to the x-axis labels and I recommend removing.